# ALDH1A3 Contributes to Radiation-Induced Inhibition of Self-Renewal and Promotes Proliferative Activity of p53-Deficient Glioblastoma Stem Cells at the Onset of Differentiation

**DOI:** 10.3390/cells13211802

**Published:** 2024-10-31

**Authors:** Andreas Müller, Bogdan Lyubarskyy, Jurij Tchoumakov, Maike Wagner, Bettina Sprang, Florian Ringel, Ella L. Kim

**Affiliations:** Laboratory for Experimental Neurooncology, Clinic for Neurosurgery, Johannes Gutenberg University Medical Centre, 55131 Mainz, Germany; mandre@students.uni-mainz.de (A.M.); blyubars@students.uni-mainz.de (B.L.); jtchouma@students.uni-mainz.de (J.T.); wmaike@students.uni-mainz.de (M.W.); bettina.sprang@unimedizin-mainz.de (B.S.); florian.ringel@unimedizin-mainz.de (F.R.)

**Keywords:** ALDH1A3, glioblastoma stem cells, radioresistance

## Abstract

ALDH1A3 is a marker for mesenchymal glioblastomas characterized by a greater degree of aggressiveness compared to other major subtypes. ADH1A3 has been implicated in the regulation of stemness and radioresistance mediated by glioblastoma stem cells. Mechanisms by which ALDH1A3 promotes malignant progression of glioblastoma remain elusive posing a challenge for rationalization of ALDH1A3 targeting in glioblastoma, and it is also unclear how ALDH1A3 regulates glioblastoma cells stemness. Usage of different models with diverse genetic backgrounds and often unknown degree of stemness is one possible reason for discrepant views on the role of ALDH1A3 in glioblastoma stem cells. This study clarifies ALDH1A3 impacts on glioblastoma stem cells by modelling ALDH1A3 expression in an otherwise invariable genetic background with consideration of the impacts of inherent plasticity and proliferative changes associated with transitions between cell states. Our main finding is that ALDH1A3 exerts cell-state dependent impact on proliferation of glioblastoma stem cells. We provide evidence that ALDH1A3 augments radiation-induced inhibition of self-renewal and promotes the proliferation of differentiated GSC progenies. Congruent effects ALDH1A3 and radiation on self-renewal and proliferation provides a framework for promoting glioblastoma growth under radiation treatment.

## 1. Introduction

Glioblastoma (GB) is the most common and most malignant form of brain tumors in adults. Despite aggressive treatment consisting of surgical resection followed by hypofractionated ionizing radiation (IR) and alkylating chemotherapy with temozolomide (TMZ), the clinical outcome of GB is poor due to a virtually inevitable recurrence after standard therapy. At the stage of recurrence, no standard therapy has been established so far, rendering recurrent GB a lethal condition [1,2,3].

The high degree of molecular and cellular heterogeneity is the biological basis of the notorious resistance of GBs to cytotoxic and targeted therapies. On the molecular level, there is a multiplicity of genomic aberrations in key oncopathways involved in the regulation of intrinsic and extrinsic responses in cancer cells [4]. With the advent of large-scale, high-throughput, next-generation sequencing methods, a more precise stratification of GBs has enabled to categorize GBs into distinct molecular subclasses. Proneural (PN), classical (Cl), and mesenchymal (Mes) GBs comprise three major subclasses that bear molecular resemblance to the various stages of developmental neurogenesis and differ in clinical outcomes [5,6]. PN GBs are more common in young patients and have a better prognosis compared with Cl or Mes GBs, with the latter type having the worst prognosis and bear characteristic traits associated with epithelial–mesenchymal transition [5]. On a cellular level, GBs are characterized by marked heterogeneity manifest as morphohistological diversity of tumor cells within the same tumor. The current biological paradigm for GB genesis and progression is centered on so-called glioblastoma stem cells (GSCs) implicated as the major source of intratumor heterogeneity and adaptability to cytotoxic insults such as those imposed by IR or non-targeted chemotherapy. Due to the intrinsic plasticity inherent to all stem cells, GSCs are capable of converting between different cellular states, characterized by distinct molecular programs and morphophenotypes. In contrast to terminal differentiation of normal stem cells, which is unidirectional and accompanied with gradual loss of the proliferative activity, cancer stem cells differentiation is aberrant and uncoupled from proliferative inhibition. In conjunction with aberrant differentiation, the high degree of plasticity renders GSCs capable of adapting to cytotoxic treatments by activating pro-survival programs that promote tumor growth. Indeed, current evidence indicates that clinically relevant cytotoxic treatments such as IR can induce a transition from the PN phenotype towards a more aggressive Mes phenotype [7] and promote an increase in tumor-propagating potential [8,9]. Bearing in mind that mesenchymal transformation is a cardinal driver for cancer progression [10], delineation of the molecular factors involved has fundamental clinical importance. The A3 isoform of aldehyde dehydrogenase 1 (ALDH1A3) has emerged as key regulator of mesenchymal transformation of GSCs [7,11]. ALDH1A3 is a member of the ALDH superfamily, comprising 19 enzymes involved in multiple processes including cell protection against oxidative stress by metabolizing toxic aldehydes, biosynthesis of retinoic acid (RA), a pleiotropic transcription factor involved in the regulation of neurogenesis, and metabolites that are crucial for central nervous system development and homeostasis [12]. In neoplasia, ALDH1A1 and ALDH1A3 isoforms have been implicated as tumor-promoting factors and universal biomarkers of cancer stem cells (CSCs), including GSCs [13,14]. Although structurally similar, ALDH1A1 and ALDH1A3 isoforms seem to be functionally non-redundant and show different expression patterns in different GB subtypes. Recent advances have enabled a more precise delineation of a relationship between individual ALDH isoforms and particular molecular subtype and clinical outcomes. Integrative molecular profiling of GB tissues indicates that elevated expression of the ALDH1A3 isoform is associated with MES GBs [7,11,15,16,17,18], whereas ALDH1A1 is more prominent in Cl GBs [19]. In accordance with profiling data from GB tissues, ALDH1A3 is also a predominant isoform in GSCs that derive from MES GBs [7,11,16,20,21,22].

While the potential diagnostic value of ALDH1A3 as a marker for MES GBs has been established, the exact role of ALDH1A3 in GSCs stemness is less clear. On the one hand, ALDH1A3 has been implicated in promoting GSCs self-renewal [7,15], but on the other hand, there is also evidence supporting the role of ALDH1A3 as a key driver of GSC mesenchymal differentiation [11,17]. Unresolved conceptual ambiguity around the role of ALDH1A3 in the regulation of GSC stemness reflects the general lack of consensus regarding the role of self-renewal in the tumor-propagating capacity of GSCs. On the one hand, self-renewal is a slow mode of division that ensures genomic integrity and error-free propagation of undifferentiated GSCs [23]. However, self-renewal may be dispensable for the dynamic tumor expansion that relies on fast-proliferating tumor cells. Indeed, it has been shown that self-renewal does not predict GSCs capacity to propagate GB growth, which is effectively driven by fast-proliferating differentiated progenies incapable of self-renewal [24]. There are also potential methodological caveats that may explain discrepant views of ALDH1A3 as either a self-renewal-promoting or differentiation-promoting factor. One is that many studies on ALDH1A3 have used conventional glioblastoma cell lines that conform with the differentiated phenotype and poorly recapitulate undifferentiated GSCs. Another potential caveat is that ALDH1A3 impacts on GSCs have been investigated only in self-renewing GSCs without consideration of their inherent plasticity, which is essential for GSC adaptability and survival under adverse conditions. This study addresses the role of cell status as a factor influencing the impacts of ALDH1A3 on GSC stemness and proliferation.

## 2. Materials and Methods

Cells and culture conditions: The human GSC lines used in this study derive from newly diagnosed GB tissues and have been extensively characterized in previous studies [8,9,25]. GSCs were cultivated in NeuroBasal medium, supplemented with the B27 component (Invitrogen Life technologies, Carlsbad, CA, USA), either in the presence (self-renewal condition) or absence of self-renewal-promoting factors bFGF (10 ng/mL) and EGF (20 ng/mL) purchased from Biochrom GmbH (Merck KGaA, Darmstadt, Germany).

Cells irradiation: GSC spheres were triturated one day before radiation using a Gulmay RS225 GS014 X-ray machine (Gulmay Medical Ltd., Camberley, UK) at a dose rate of 1 Gy/min. The radiation regimen consisted of five daily fractions of 2 Gy given consecutively. Immediately after irradiation, cells were placed back to the incubator for recovery and propagation until the next round of radiation treatment. Except for irradiation, control cultures were handled identically in parallel. After the completion of radiation treatment, radiated or mock-treated GSCs were expanded and used for comparative analyses.

Cell-based assays: Self-renewal was determined by using the extreme limiting dilution assay (ELDA) [26], which enables to quantitatively assess self-renewal by determining the proportion of cells that are capable of dividing while maintaining the undifferentiated state and multipotency. In brief, GSC spheres were dissociated via a combined enzymatic–mechanic trituration procedure using Trypsin (Gibco, 25300-054, via Fischer Scientific GmbH, im Heiligen Feld, Germany) diluted at a 1:1 ratio in NeuroBasal medium supplemented with B27 component, bFGF (10 ng/mL), and EGF (20 ng/mL). Cells were resuspended in NeuroBasal + B27 medium and analyzed by trypan blue exclusion test to assure that cell suspensions had the required viability (≥95%). Initial cell suspensions were subjected to serial dilutions in NeuroBasal + B27 medium supplemented with bFGF/EGF to obtain secondary suspensions with defined concentrations of cells (50.0, 25.0, 12.5, 6.25, 3.125, and 1.56 cells/mL). Serially diluted suspensions were distributed into 24 wells plates at 1 mL per well with at least 30 replicate wells for each cell concentration. Plates with cells were placed in an incubator and examined under the microscope for the appearance of GSC spheres starting at four weeks after cells plating. Number of sphere-containing wells was determined by microscopical examination and entered as the “response” value, along with the overall number of wells tested for each cell concentration (“tested”) and corresponding cell concentration (“dose”), into ELDA software (https://bioinf.wehi.edu.au/software/elda/ (assessed on 15 April 2024)) for value-based algorithmic calculations of stem cell frequency. For growth curve analysis, cells were plated in 6-well plates at a density of 2 × 10^4^ cells per well and incubated for indicated times. Three wells were used as technical replicates for each timepoint. At each timepoint, cells were collected and subjected to trituration to prepare single-cell suspensions. Viable cell counts were determined by using the trypan blue exclusion method. Data from three replicates were pooled and plotted on a linear scale. For the colony formation assay, cells were plated in 6-well plates at a density of 1 × 10^3^ cells per well and incubated for 5–6 weeks. Three wells were used as technical replicates for each timepoint. After incubation, the medium was removed and replaced with 2 mL PBS to wash the cells. After the wash step, cells were fixed with a methanol/acetic acid mix (3:1) for 5 min at room temperature and stained with crystal violet (0.5% in methanol) for 15 min at room temperature. Stained cells were washed with water, dried, and counted under a microscope. For differentiation, GSCs were plated on ornithine-coated glass coverslips at 3000 cells/coverslip in NeuroBasal/B27 medium without bFGF or EGF and incubated for 10 days. For immunofluorescence staining, GSCs were fixed with 4% paraformaldehyde, washed with PBS, and incubated in blocking solution (0.1% Triton X-100, 1% bovine serum albumin) for 1 h. After blocking, cells were stained overnight at +4 °C with primary antibodies diluted in blocking solution. Primary antibodies used in this study include α-Ki67 (Abcam, Cambridge, UK; ab16667), α-nestin (Abcam, Cambridge, UK; ab22035), α-GFAP (Dako Z0334), and α-phospho-Histone3 S10 (Cell Signaling Technology, Inc., Danvers, MA, USA; #9701). Secondary antibodies were goat α-mouse Alexa Fluor 488 (Invitrogen, Waltham, MA, USA; A-11001, 1:10,000) or goat α-rabbit Alexa Fluor 555 (Invitrogen, Waltham, MA, USA; A-21429).

Immunohistochemistry: Whole brains extracted from GSC-implanted mice were fixed in 4% paraformaldehyde and embedded into paraffin blocks using a standard paraformaldehyde–paraffin embedding (FFPE) protocol. Three-micrometer sections were prepared from FFPE mouse brains, transferred to glass slides, and stained with antibodies specific for human nestin (Invitrogen, Waltham, MA, USA; PA5-82905) as previously described [8,9]. 

Lentivirus-mediated transfer: GSC-1095 cells were transduced with pLenti-C-mGFP-Puro tagged ORF ALDH1A3 or pLenti-C-mGFP-P2A vectors purchased from OriGene Technologies, Inc. (Rockville, MD, USA; Cat. Numbers RC209656L4 and PS100093, respectively) using an optimized protocol for glioblastoma stem cells [27]. Transduced clones were selected by resistance to puromycin, verified by flow cytometry for GFP and expanded for downstream analyses under self-renewal-promoting culture conditions (NeuroBasal medium supplemented with the B27 component (Invitrogen Life technologies, Carlsbad, CA, USA)), bFGF, and EGF (Biochrom GmbH, Merck KGaA, Darmstadt, Germany).

Western blot: Cells were lysed in SDS lysis buffer (1% SDS, 1 mM Tris, 1 mM EDTA, pH 8,0) supplemented with a protease inhibitor cocktail (cOmplete^TM^, Sigma-Aldrich, St. Louis, MO, USA) for 10 min at 95 °C and subjected to ultrasound sonification using an Ultrasonicator (Bandelin Sonopuls, Berlin, Germany). After sonification, cell lysates were cleared by centrifugation at 14,000 rpm for 15 min at +4 °C. Protein concentration was determined by using a NanoDrop spectrophotometer (NanoDrop Technologies, Wilmington, DE, USA). Electrophoretic protein separation was achieved by using precast gel mini-PROTEAN^®^ TGX™ (Bio-Rad) followed by protein transfer on a PVDF membrane (ThermoFischer Scientific Inc., Waltham, MA, USA). Antibodies used were anti-ALDH1A3 (Invitrogen, Waltham, MA, USA; PA5-29188), anti-CD133 (Miltenyi Biotec, Bergisch Gladbach, North Rhine-Westphalia, Germany; 130-092-395), anti-β-actin (Santa Cruz, Santa Cruz, CA, USA; Sc-47778), anti-H2B (Cell Signaling, Danvers, MA, USA; #2934), anti- PDGF Receptor α (Cell Signaling, Danvers, MA, USA; #5241), and anti- PDGF Receptor β (Cell Signaling, Danvers, MA, USA; # 3169). Signal intensity was quantified by densitometry using ImageJ version 1.x (https://imagej.nih.gov, accessed on 1 March 2024).

Statistical analysis: Data analysis was performed by using Student’s *t*-test and presented as mean ± SD. A *p* value < 0.05 was considered statistically significant.

## 3. Results

### 3.1. ALDH1A3 Expression in GSCs Differing for the Status of Self-Renewal Factor CD133

Associations between expression levels of putative biomarkers and distinct molecular subtypes are mostly based on the data from transcriptional or genomic profiling. To gain insight into the expression patterns of ALDH1A3 at the protein level, steady-state levels of the ALDH1A3 protein were evaluated by Western blot along with stemness markers CD133, PDGFRA, or PDGFRB. To gain insight into the expression patterns of ALDH1A3 at the protein level, steady-state levels of the ALDH1A3 protein were evaluated by Western blot along with stemness markers CD133, PDGFRA, or PDGFRB. The results showed that ALDH1A3 expression varies considerably across heterologous GSCs derived from either newly diagnosed or recurrent glioblastoma (ndGB or recGB, respectively, Figure 1).

Notably, ALDH1A3 levels do not parallel those of CD133, PDGFRA, or PDGFRB. In fact, CD133 and ALDH1A3 abundance appear to correlate inversely with each other (Figure 2).

Furthermore, glioblastoma cell line U87, predominantly comprising differentiated cells, has higher levels of ALDH1A3 than any self-renewal competent GSC analyzed in this study (Figure 2). Considering that CD133, PDFGRA, and PDGFRB factors are essential for GSC self-renewal [28,29,30,31,32], the lack of correlation between the levels of these factors and ALDH1A3 prompted us to check whether ALDH1A3 abundance correlates with self-renewal capacity. To that end, we employed the ELDA assay, which enables the quantitative assessment of self-renewal by determining the proportion of cells that are capable of dividing while maintaining an undifferentiated state [26]. In brief, self-renewal capacity was compared between GSCs with either low (GSC-649, GSC-619) or high (GSC-696) levels of ALDH1A3, by ELDA. The results showed that ALDH1A3 abundance does not reflect the degree of self-renewal. For example, GSC-619, GSC-649, and GSC-696 lines, differing markedly in their levels of ALDH1A3 (Figure 1), have comparable levels of self-renewal capacity (Figure 3).

### 3.2. Modelling Differential Expression of ALDH1A3 in GSCs with Invariable Molecular Background

In light of the marked variations in the expression of stemness-associated factors across heterologous GSCs (Figure 1 and Figure 2), it is difficult to distinguish the impacts of ALDH1A3 on GSC stemness. In order to clarify this question unequivocally, we sought to generate an experimental model that would enable the modulation of ALDH1A3 levels against an otherwise invariable molecular background. To that end, we used GSC line GSC-1095, which has high self-renewal capacity, is capable of differentiation, and recapitulates histopathological and clinical characteristics of human GB such as highly invasive tumor phenotypes, intratumoral phenotypic heterogeneity (Figure 4), and inherent radioresistance [8].

In order to obtain isogenic models with differential expression of ALDH1A3 GSC-1095, cells were stably transduced with lentiviral vectors coding for either GFP-fused human ALDH1A3 or GFP only. Transduced clones (conditionally termed as “1095^ALDH1A3^” or “1095^vector^”, respectively) were validated at the mRNA and protein levels (Figure 5) and used in parallel in comparative downstream analyses.

### 3.3. ALDH1A3 Impacts on GSC Self-Renewal

The apparent lack of correlation between ALDH1A3 abundance and self-renewal capacity (Figure 1, Figure 2 and Figure 3) urge further clarification regarding the impact of ALDH1A3 on self-renewal. It has been proposed that ALDH1A3’s impacts on GSCs stemness might be associated with radiation [27]. With this in mind, we compared self-renewal between GSCs that differ in their levels of ALDH1A3 (Figure 5) and have been either untreated or irradiated, by ELDA [26]. Parallel ELDA assessments revealed no considerable difference in self-renewal capacity between non-radiated 1095^ALDH1A^ and 1095^vector^ GSCs (Figure 6A).

With keeping in mind that ALDH1A3’s impacts on GSCs stemness may become manifest in the context of radiation as proposed by Mao et al. [7] self-renewal was also compared between GSCs that have been subjected to clinically relevant doses of IR (2.5 Gy daily fractions applied for five consecutive days). The results showed that IR-treated 1095^ALDH1A3^ and 1095^vector^ GSCs (hereafter termed as “1095^ALDH1A3^_IR” and “1095^vector^_IR”, respectively) differ significantly in their capacity to self-renew. Indeed, ALDH1A3-overexpressing GSCs proved to have a significantly lower frequency of self-renewing cells (1 out of 18) compared to their counterpart 1095^vector^_IR (one out of 7) (Figure 6B). Notably, self-renewal activity of either 1095^ALDH1A3^_IR or 1095^vector^_IR GSCs is lower than that of their non-radiated counterparts, indicating that IR treatment per se has an inhibiting effect on self-renewal. However, the extent of self-renewal inhibition by IR is profoundly greater in 1095^ALDH1A3^_IR GSCs compared to 1095^vector^_IR GSCs. Collectively, ELDA results indicate that high abundance of ALDH1A3 is ineffective in inhibiting self-renewal in non-radiated GSCs (Figure 6A), but it does augment inhibition of self-renewal triggered by IR (Figure 6B).

### 3.4. Impacts of ALDH1A3 on Differentiation Efficiency

Given that exit from of self-renewal is accompanied by gradual transitions to more differentiated states, we asked if a reduction in self-renewal ability after radiation (Figure 6B) may be associated with the increase in differentiation potential. To address the possibility, differentiation efficiency was estimated by determining the percentage of cells that acquire the GFAP+ phenotype, which marks the onset of differentiation in GSC-1095 cells (Figure 4A). GFAP assessments showed that the percentage of GFAP+ cells appearing under differentiation-inducing conditions (“bFGF−/EGF−”) does not differ between 1095^vector^ and 1095^ALDH1A3^ GSCs (Figure 7, grey bars), indicating that differentiation efficiency is not influenced by ALDH1A3 in non-radiated GSCs.

The fraction of GFAP+ cells is even lesser (modestly but significantly) in 1095^ALDH1A3^_IRs compared to 1095^vector^ _IRs (red bars). These results indicate ALDH1A3 does not increase differentiation in either naive or radiated GSCs. On the contrary, the differentiation efficiency of radiated GSCs may even be slightly reduced in GSCs possessing presence of high ALDH1A3 levels.

### 3.5. Impacts of ALDH1A3 on Proliferation

Cell fate changes are accompanied by changes in the proliferative capacity of stem cells. Our finding that ALDH1A3 has a strong impact on self-renewal in the context of radiation (Figure 6B) prompted us to test if ALDH1A3 has an effect on the proliferative activity of radiated GSCs. To that end, proliferative activity was evaluated by assessing the marker of proliferation Ki67 and mitosis marker phospho-Histone 3 (H3S10). Comparative assessments revealed that ALDH1A3 abundance has no significant influence on the proliferation of either non-radiated or radiated GSCs in a state of self-renewal (Figure 8A). Although proliferation activity is somewhat reduced in radiated GSCs compared to non-radiated GSCs, the reduction is not significantly related to the difference in ALDH1A3 levels in the state of self-renewal. However, comparison of proliferative capacity under differentiation-inducing conditions (Figure 8B) revealed profound differences in the proliferative capacity between 1095^ALDH1A3^ and 1095^vector^ GSCs. While both 1095^vector^ and 1095^ALDH1A3^ GSCs undergo a reduction in proliferative activity upon differentiation induction, the degree of proliferative decline differs markedly between the two lines, with the proportion of proliferating cells being nearly twice as large in 1095^ALDH1A3^ GSCs compared to 1095^vector^ GSCs (Figure 8B,C). Proliferative changes manifested by radiated GSCs under differentiation-inducing conditions differ strikingly from those in naive GSCs. 1095^ALDH1A3^_IR and 1095^vector^_IR GSCs not only fail to cease proliferation after the exit from self-renewal but, quite on the contrary, they have an even higher proliferative activity than their non-radiated counterparts (Figure 8C). ALDH1A3 impact on proliferation appears to be more profound in non-radiated GSCs than in radiated GSCs (Figure 8C). Although 1095^ALDH1A3^_IRs show a trend towards increased proliferation compared to 1095^vector^_IR GSCs, the difference is not significant. These results demonstrate that ALDH1A3 exerts differential impacts on GSCs depending on their fate. While having no significant impact on proliferation of undifferentiated GSCs, ALDH1A3 promotes proliferation of differentiated progenies. Concordant with this conclusion, cells that actively proliferate under differentiation-inducing conditions have the morphology of differentiated cells and express GFAP (Figure 9A) in contrast to nearly exclusively GFAP- phenotype of undifferentiated cells proliferating under self-renewal conditions (Figure 9B).

Further supporting the conclusion that ALDH1A3 promotes the proliferation of differentiated progenies, a proportion of double-positive H3S10+/GFAP+ cells is significantly higher in 1095^ALDH1A3^ subline compared to 1095^vector^ subline (Figure 9C)

### 3.6. ALDH1A3 Expression Promotes the Clonogenic Potential and Growth in Differentiated Progenies

Our finding that ALDH1A3 expression promotes proliferation at the onset of differentiation (Figure 8C and Figure 9C) raises the possibility that ALDH1A3 has an impact on clonogenic capacity. To address this possibility, we evaluated the colony formation ability of 1095^vector^ and 1095^ALDH1A3^ cells. While both 1095^vector^ and 1095^ALDH1A3^ cells proved capable of forming colonies in a differentiated state, 1095^ALDH1A3^ colonies appear to be more abundant and generally larger compared to those formed by 1095^vector^ cells (Figure 10A).

To validate this observation, we applied a quantitative approach that enables the determination of the size of colonies (Appendix A). The results showed that both the range and size of most frequent colonies formed by 1095^ALDH1A3^ cells are markedly greater than those of 1095^vector^ colonies (Figure 10B), indicating that ALDH1A3 enhances clonogenic potential. To corroborate this conclusion, clonogenic capacity was evaluated by colony forming assay. The results showed that the clonogenic capacity of 1095^ALDH1A3^ cells is significantly higher than that of 1095^vector^ cells (Figure 10C, grey bars). Furthermore, the clonogenicity of 1095^ALDH1A3^ cells is not only undiminished but is even further increased upon radiation (red bars). Notably, radiation also increases the clonogenic capacity of 1095^vector^ cells, indicating that the clonogenicity-stimulating effects of radiation do not rely exclusively on ALDH1A3. Collectively, the results of the clonogenicity analyses predict that ALDH1A3 promotes growth in differentiated GSC progenies. To test this possibility, we compared the growth dynamics between 1095^ALDH1A3^ and 1095^vector^ cells. The results showed that 1095^ALDH1A3^ cells grow significantly faster compared to 1095^vector^ cells (Figure 10D, upper graph). Notably, slow but steady growth under differentiation-inducing conditions is preceded by a sharp decline in the 1095^vector^ but not 1095^ALDH1A3^ population, which is consistent with a profound difference between the two lines, in the degree of proliferative inhibition at the onset of differentiation (2.7- vs. 1.6-fold, respectively, Figure 8C). The growth dynamics of 1095^vector^_IR or 1095^ALDH1A3^_IR populations also mirror the effects of radiation on proliferation (Figure 8C), with both sublines having much faster growth rates than their non-radiated counterparts (Figure 10D, bottom). Moreover, confirming the trend suggested by assessments of proliferative activity (Figure 8B), 1095^ALDH1A3^_IR cells proved to have a much faster population doubling time than 1095^vector^_IR line (32 days vs. 24 days, respectively).

## 4. Discussion

This study reports on the role of ALDH1A3 in GSCs stemness and proliferation. Our investigations reveal a previously unsuspected role of ALDH1A3 in the regulation of GSCs self-renewal and cell state-dependent proliferation. In contrast with the view that ALDH1A3 has a generally promoting effect on GSCs self-renewal [7,15] we provide evidence that ALDH1A3 not only does not enhance self-renewal in p53-deficient GSCs but instead strongly inhibits self-renewal in a radiation-dependent manner. Inhibition of self-renewal appears to be an inducible activity of ALDH1A3 because it becomes manifest in conjunction with radiation treatment but not in non-radiated GSCs. Our results show that increased expression of ALDH1A3 is associated with increased proliferation of GSCs that have exited from self-renewal and committed to differentiation. Considering that stem cells transition from self-renewal to more differentiated states is normally accompanied by cessation of proliferation, our findings suggest the role of ALDH1A3 in the uncoupling between differentiation-promoting and proliferation-inhibiting signaling. That ALDH1A3 impacts on GSC proliferation are both cell state- and radiation-dependent suggests that there might be an interconnection between ALDH1A3 and other factors involved in the regulation of the balance between self-renewal, differentiation, and proliferation. In this regard, it should be noted that the GSC models used in our study lack p53, which is an important regulator of self-renewal and differentiation in neural stem cells and GSCs, especially in the context of radiation [33,34,35,36]. Further underscoring the potential interplay between ALDH1A3 and p53 in the regulation of GSCs stemness, p53 is a transcriptional activator of the *aldh1a3* gene [37,38,39]. It would be interesting to test if and how the effects of ALDH1A3 uncovered in our study using p53-null GSCs may be modulated depending on the status of p53.

Our investigations provide new insights into the impacts of radiation on GSCs stemness and proliferation. While concordant with the general view that exit from self-renewal and induction of differentiation is a protective response to radiation in neural stem cells [40,41], our data indicate that radiation-induced inhibition of self-renewal does not necessarily lead to increased differentiation in GSCs but instead it can augment the proliferative capacity of differentiated progenies.

Usage of isogenic models in our study enabled us to establish unequivocally that ALDH1A3 and IR can boost proliferation of differentiated progenies independently from each other. That the combination of both factors does not generate a significant increment in GSC proliferation at the onset of differentiation suggests that ALDH1A3 and IR promote differentiation-associated proliferation via non-overlapping mechanisms. Considering that the proliferative boost induced by radiation is profoundly greater compared to that caused by ALDH1A3 overexpression alone, it is possible that ALDH1A3 effects may be overshadowed by a more potent impact exerted by IR. Nevertheless, our data indicate that ALDH1A3 does cooperate with IR in promoting growth rates and increasing clonogenicity in differentiated progenies.

Our findings support the view that differentiated progenies of GSCs make an important if not major contribution in GB propagation [24], especially after radiation. We provide evidence for previously unsuspected roles of ALDH1A3 in overcoming proliferative inhibition at the onset of GSC differentiation. Considering that differentiated tumor cells constitute the major population in GB tissues, the ability of ALDH1A3 to boost proliferation in differentiated progenies of GSCs further underscores the potential merit of ALDH1A3 inhibition as a therapeutic approach.

Our finding that ALDH1A3 promotes proliferation and growth potential in differentiated progenies rather than undifferentiated GSCs may explain discrepant findings from previous studies in which the degree of differentiation was not considered. It should be emphasized, though, that ALDH1A3 impacts uncovered in p53-null GSCs may not be a universal rule but a distinctive characteristic of GSCs/GSC progenies that lack p53 activities. Considering that p53 plays an important role in both cell fate determination and radiation response, it cannot be excluded that ALDH1A3 impacts may differ against p53-proficient and p53-deficient backgrounds. While further investigations are needed to clarify the possibility, our finding that ALDH1A3 activities are modulated during cell state transitions provides a conceptual framework for further investigations on the complex regulation of ALDH1A3-mediated responses in GSCs.

## 5. Conclusions

Based on the results of our investigations, we conclude that ALDH1A3’s effects on GSCs are dependent on cell state and associated with responses induced by radiation. ALDH1A3 overexpression in conjunction with radiation leads to a reduction in self-renewal and increase in the proliferative and clonogenic capacity of differentiated progenies. The cooperative effects of ALDH1A3 and radiation in boosting proliferation of differentiated progenies underscore the potential merit of ALDH1A3 inhibition as an approach for impeding glioblastoma radioresistance.

## Figures and Tables

**Figure 1 cells-13-01802-f001:**
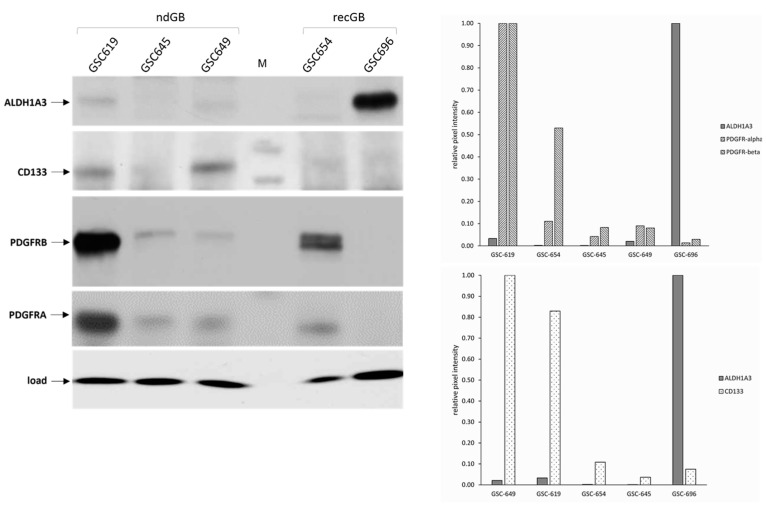
Western blot assessments of endogenous ALDH1A3 and self-renewal-promoting factors CD133, PDGFRA, and PDGFRB in heterologous GSCs. “ndGB”: newly diagnosed GB. “recGB”: recurrent GB. “load”: histone2B used as protein loading control. “M”: protein marker. Graphs show quantification of ALDH1A3, PDFGRA/B, and CD133 signals.

**Figure 2 cells-13-01802-f002:**
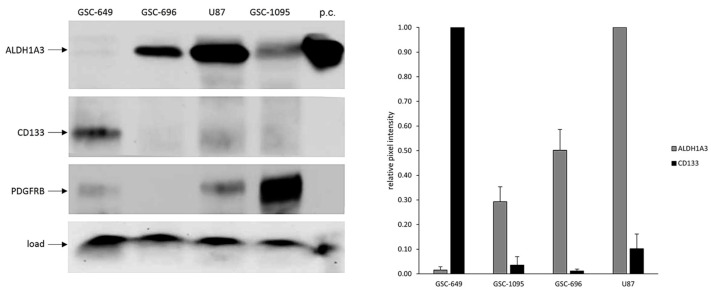
Comparative assessments of endogenous ALDH1A3 in CD133 + GSCs, CD133 – GSCs, and non-stem glioblastoma cell line U87 used as a reference. GSC-1095 is a p53-deficient line that derives from newly diagnosed GB and has been previously characterized with respect to the tumor-initiating capacity and intrinsic radioresistance [8]. “p.c.”: recombinant ALDH1A3 protein used as antibody specificity control. “load”: histone2B used as a control for equal protein loading. Graph shows quantification of ALDH1A3 and CD133 signals. Data from three independent experiments.

**Figure 3 cells-13-01802-f003:**
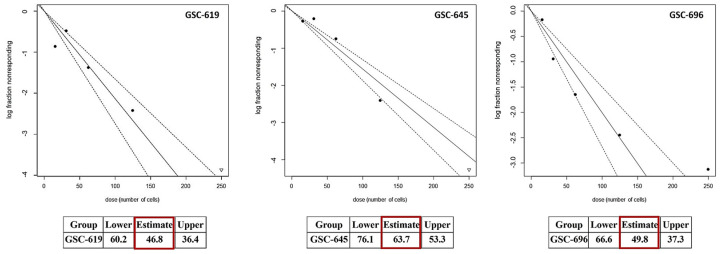
Quantitative assessments of self-renewal by ELDA. Graphs show the data from three independent experiments. Tables below show estimated frequency of self-renewing cells (red-framed) and confidence intervals for 1/(stem cell frequency).

**Figure 4 cells-13-01802-f004:**
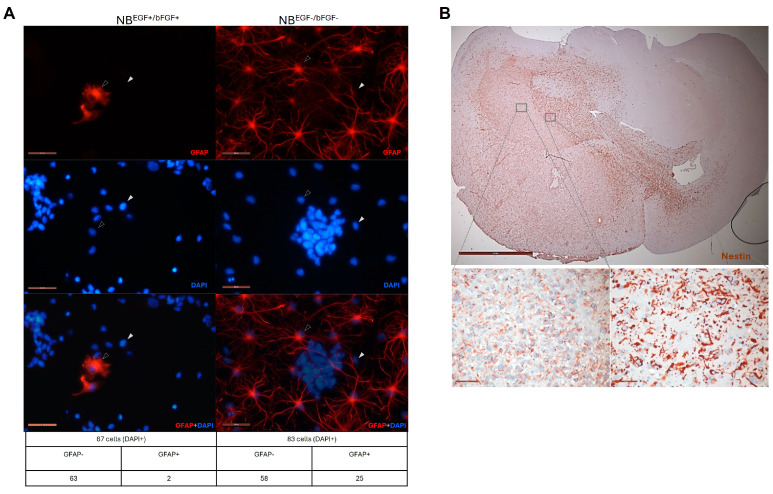
Stemness properties of GSC-1095. (**A**) Cell fate plasticity manifest in morphological changes and induction of the astrocytic lineage marker GFAP under differentiation-inducing culture conditions (withdrawal of self-renewal factors bFGF and EGF, “NB^bFGF−/EGF−^”). (**A**) Immunofluorescence staining for the astrocyte differentiation marker glial fibrillary acidic protein (GFAP, red). Nuclei counterstaining by DAPI (blue). Magnification 40× (scale bar = 50 µm). Representative undifferentiated (GFAP−) or differentiated (GFAP+) cells are indicated by solid or empty arrowheads, respectively. Numbers of GFAP−/DAPI+ and GFAP+/DAPI+ cells are indicated in the table below the image. (**B**) Immunohistochemical staining of GSC-1095 xenograft tumor for a putative neural stem cell marker nestin (brown staining) using antibody specific for human nestin. Top image: Entire brain coronal section (magnification 1.6×, scale bar = 2 mm). Tumor occupies nearly all of the hemisphere and spreads into the contralateral hemisphere. Bottom images: Enlarged images of tumor regions demarcated by grey squares in the top part. Intratumoral phenotypic heterogeneity manifest in variable expression of nestin across the tumor (magnification 40×, scale bars = 50 µm).

**Figure 5 cells-13-01802-f005:**
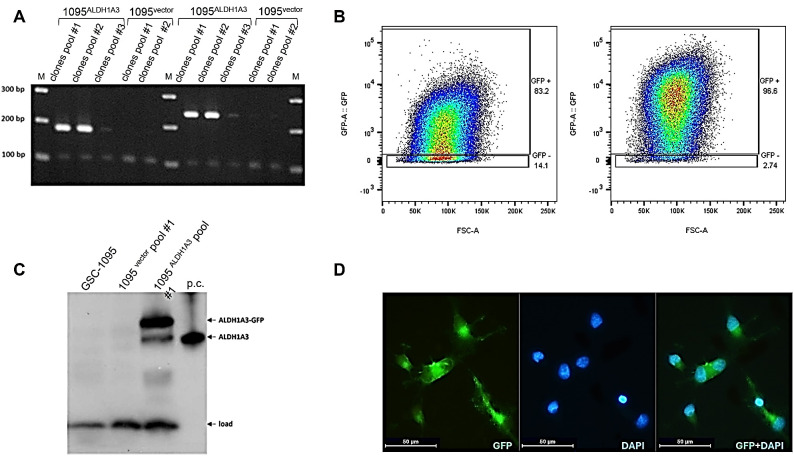
GSC model for differential expression of ALDH1A3. (**A**) RT-PCR validation of ALDH1A3 transcripts 1 and 2. Pools of clones generated at a variable multiplicity of infection with ALDH1A3-coding or “empty” vectors (1095^ALDH1A3^ and 1095^vector^, respectively). A 100 bp band corresponds to the p21 transcript co-amplified in all RP-PCR reactions as an internal control. “M”: 100 bp DNA ladder. (**B**) Flow cytometric assessments of ALDH1A3-GFP or GFP expression in cell pools used for comparative investigations. (**C**) Western blot validation of the ALDH1A3-GFP protein in 1095^ALDH1A3^ cells. “p.c.”: recombinant ALDH1A3 protein used as positive control. (**D**) Immunofluorescence staining of 1095^ALDH1A3^ cells using anti-GFP antibody (left image). Nuclear counterstaining by DAPI (middle image). Merged image (right image) shows a predominantly cytosolic localization of ALDH1A3-GFP. Magnification 40×, scale bar = 50 µm.

**Figure 6 cells-13-01802-f006:**
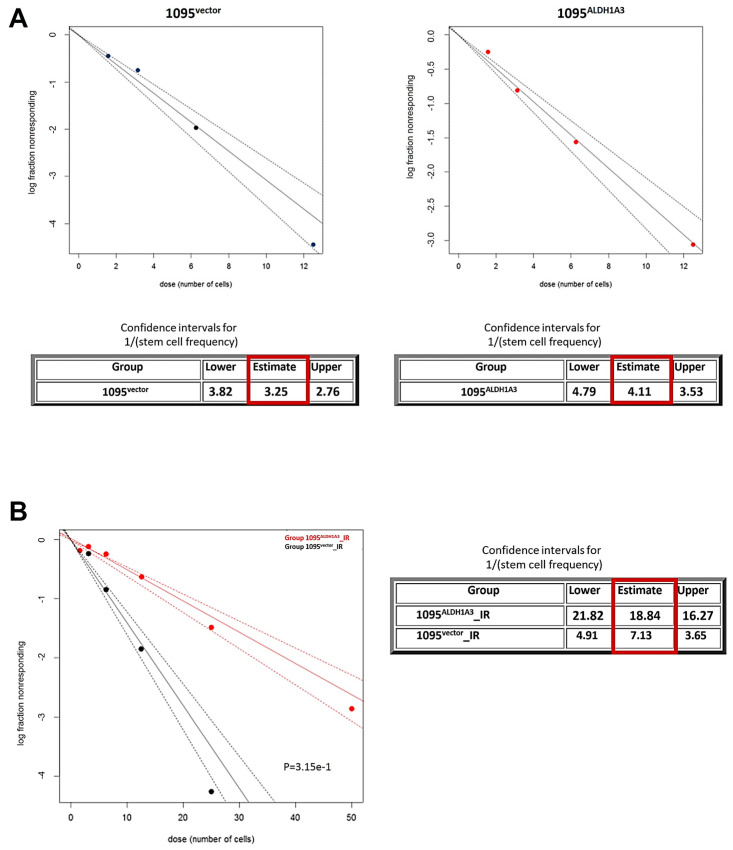
Comparative assessments of self-renewing capacity by ELDA. (**A**) Estimation of self-renewal in non-radiated GSCs 1095^ALDH1A3^ and 1095^vector^. (**B**) Estimation of self-renewal in radiation-treated GSCs 1095^ALDH1A3^_IR and 1095^vector^_IR. Graphs show the data from three independent experiments. Tables below show estimated frequency of self-renewing cells (red-framed) and confidence intervals for 1/(stem cell frequency).

**Figure 7 cells-13-01802-f007:**
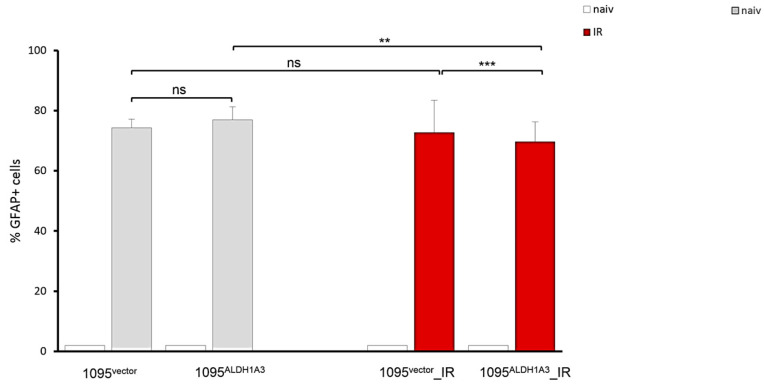
Comparative assessments of differentiation. Proportion of GFAP+ cells detected by IFS under differentiation-inducing condition (bFGF−/EGF−) in non-radiated (grey bars) or radiation-treated (red bars) GSCs. Total population of at least 500 cells was analyzed under each condition. Mean ± SEM of triplicate coverslips from three independent repeats. “ns”: not significant. **, *p* < 0.01. ***, *p* < 0.0005. For comparison, percentage of rare GFAP+ cells comprising less than 2 percent under self-renewal-promoting condition (“bFGF+/EGF+”) is also shown (white bars).

**Figure 8 cells-13-01802-f008:**
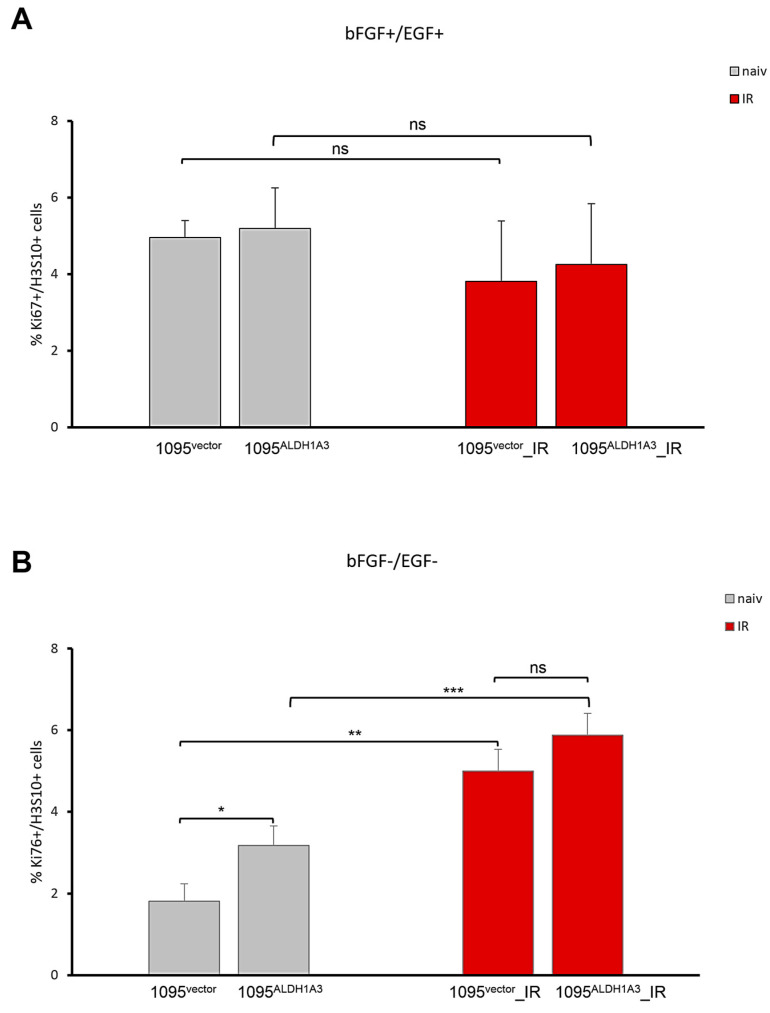
Comparative assessments of the proliferation activity. IFS data for H3Ser10+ cells. Mean ± SEM of triplicate coverslips from three independent repeats. (**A**) Self-renewal condition. (**B**) Differentiation-inducing condition. “ns”: not significant. *, *p* < 0.05, **, *p* < 0.01, ***, *p* < 0.001. (**C**) Compilation of H3S10 values for all lines.

**Figure 9 cells-13-01802-f009:**
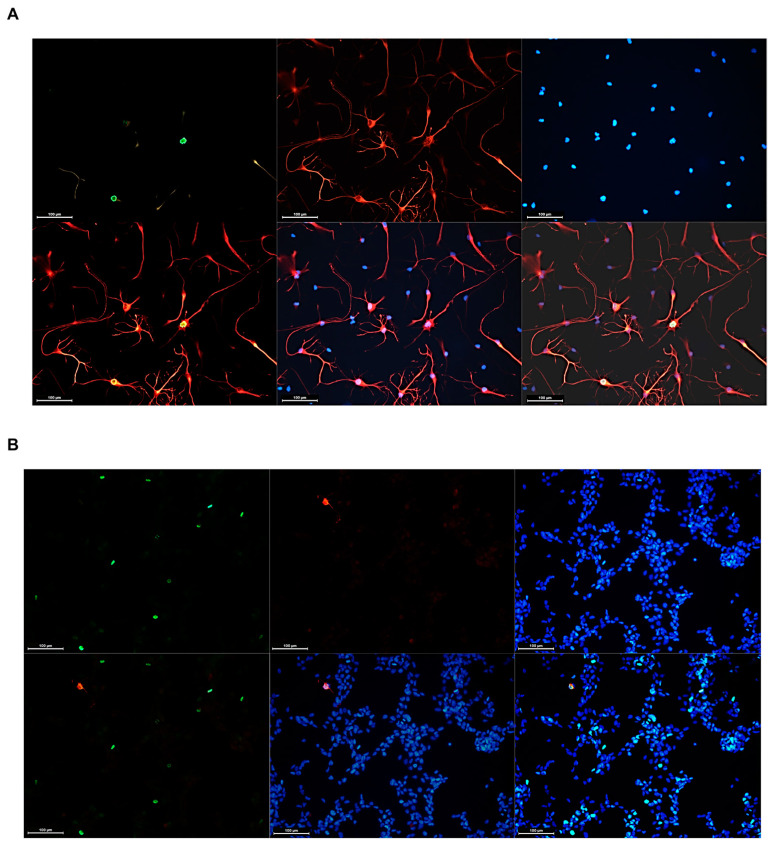
GFAP and H3S10 patterns in differentiated or self-renewing states. Immunofluorescence staining with anti-GFAP and anti-H3S10 antibodies. (**A**) GFAP+/H3S10+ phenotype of cells proliferating under differentiation-inducing condition. Upper panels, representative images of cells expressing H3S10 (green signal, fluorescence marker Alexa Fluor 488) and GFAP (red signal, fluorescence marker Alexa Fluor 555). Nuclear counterstaining by DAPI (blue). Bottom panels, merged images of double-positive GFAP+H3S10+ (**left**), GFAP+DAPI+ (**middle**), or triple-positive GFAP+H3S10+DAPI+ (**right**) cells. (**B**) GFAP-/H3S10+ phenotype associated with proliferation under self-renewal condition. Upper panels, staining patterns for H3S10 (green) or GFAP (red). Nuclear counterstaining with DAPI (blue). Bottom panels, merged images. Magnification 20×, scale bars = 100 µm. (**C**) Quantification of GFAP-/HSP3S10+ and GFAP+/H3S10+ cells in 1095^ALDH1A3^ and 1095^vector^ cells. Mean ± SEM of triplicate coverslips from three independent experiments *, *p* < 0.05.

**Figure 10 cells-13-01802-f010:**
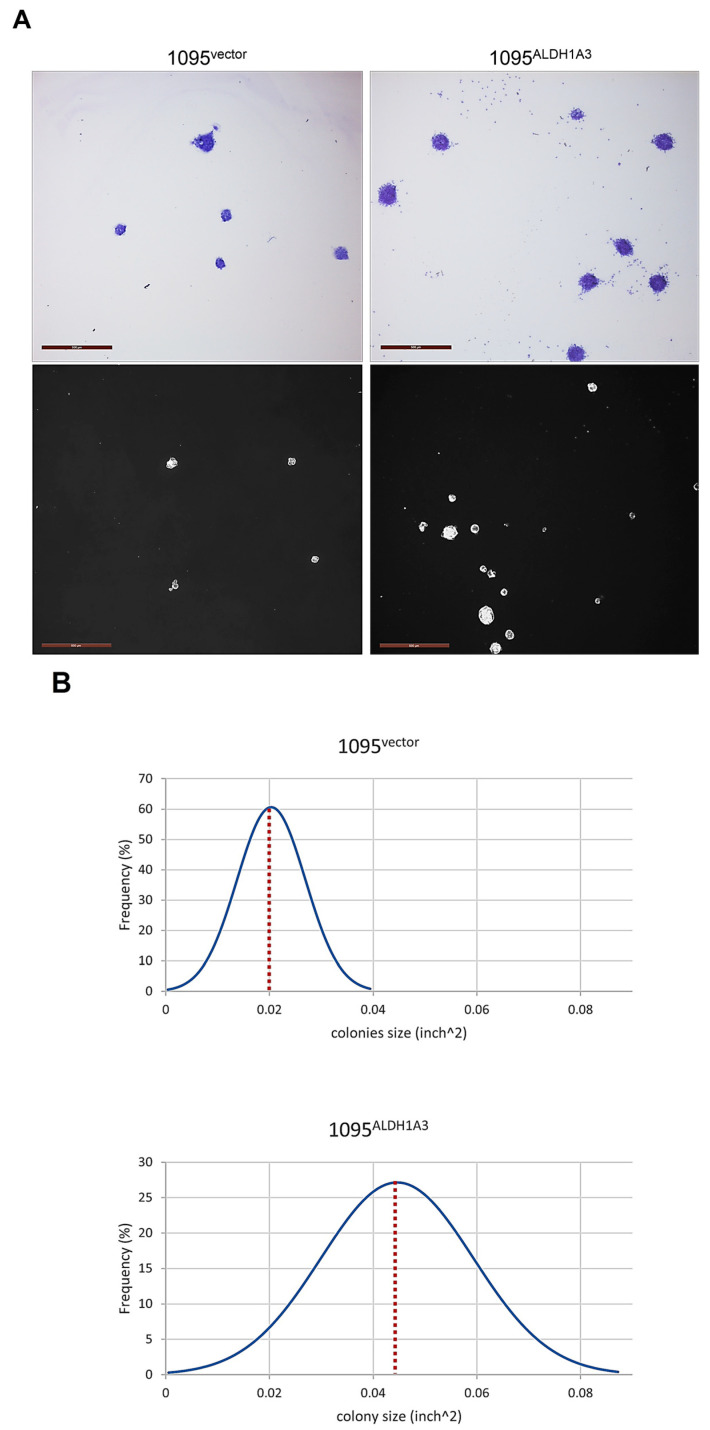
Effects of ALDH1A3 on clonogenic potential and growth rate. (**A**) Representative colonies formed by 1095^vector^ or 1095^ALDH1A3^ cells. Upper panels: colonies after fixation and staining with crystal violet. Lower panels: live colonies photographed under native condition. Magnification 5×, scale = 500 µm. (**B**) Evaluation of clonogenicity by size of colonies. Bell curves summarize the data from three independent experiments performed in triplicate. At least 100 colonies in total (30–40 colonies from each experiment) were analyzed. (**C**) Evaluation of clonogenicity by the colony formation assay. Mean ± SEM of three independent experiments. “ns”: not significant. *, *p* < 0.05; **, *p* < 0.01; ***, *p* < 0.0005. (**D**) Growth rates under differentiation-inducing condition (bFGF−/EGF−). Upper graph: parallel analyses of 1095^vector^ and 1095^ALDH1A3^ cultures. Lower graph: parallel analyses of 1095^vector_IR^ or 1095^ALDH1A3_IR^ cultures. Growth curves are representative of three independent experiments.

## Data Availability

The original contributions presented in the study are included in the article/Appendix A, further inquiries can be directed to the corresponding author.

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
