# Peer review of "ALDH1A3 Contributes to Radiation-Induced Inhibition of Self-Renewal and Promotes Proliferative Activity of p53-Deficient Glioblastoma Stem Cells at the Onset of Differentiation"

_cells, 2024, doi:10.3390/cells13211802_

Round 1

Reviewer 1 Report (Previous Reviewer 1)

Comments and Suggestions for Authors

Revision:  Müller et ail: ALDH1A3 contributes to radiation-induced inhibition of self-renewal…

In the current revision authors did not demonstrate reproducibility in additional cell line leaving conclusion speculative. Fig.2 shows that GSC 1095 has intermedium level of ALH1A3 expression. Therefore, this cell line can be used for overexpression and knockdown or knockout. Additional cell lines are also available to use in overexpression/KD/KO studies. Analysis of p53 status should not be difficult to do. Inability to generate cells with sufficient target KD is not an excuse.

Authors did not show if target expression levels correlate with tumor grade or subtype or survival.

Authors found excuse not to perform in vivo experiment. Currently characterized and fast growing GSC lines available from different laboratories.

Authors corrected only “Other comments” section of the review.

Author Response

Reviewer: In the current revision authors did not demonstrate reproducibility in additional cell line leaving conclusion speculative.

Response:

  1. This claim is unjustified as it completely ignores the fact that a substantial part of our study is comprised of experimental data obtained with different GSC lines (data shown in Figs. 1,2 and 3). Importantly, our data generated in different cell lines consistently reproduce the phenomenon investigated in our study, namely the lack of correlation between ALDH1A3 expression and self-renewal. These datasets constitute an essential part of the study (three out of ten figures, Figs. 1,2 and 3) and provide the rationale for conducting a more in-depth mechanistic investigations, which were then conducted with 1095-GSC line.
  2. It is unclear what is meant by lack of “reproducibility", does the reviewer refer to technical reproducibility or conceptual reproducibility referring to replicability of study’s conclusions? In terms of technical reproducibility defined as consistency of measurements, our study fulfils all the criteria such as repeated experiments (at least three independent experiments in triplicate), provision of detailed methods as well as usage of different experimental approaches that all yielded consistent results. In terms of conceptual reproducibility, its essence is in validation by independent studies. According with the generally accepted consensus view formulated by National Academies of Sciences, Engineering, and Medicine “Reproducibility refers to the ability of researchers to duplicate the results of a prior study using the same materials and procedures as were used by the original investigator” ("Reproducibility and Replicability in Science" consensus report 2019 https://doi.org/10.17226/25303).  We would like to stress that validation of complex biological effects requires a prior knowledge about specific factors that are relevant for a particular phenomenon/effect. In our previous response, we have addressed this aspect in great detail. As our arguments have (unfortunately) been uncommented by the reviewer we repeat them here again:  “…testing the replicability of any biological effect can be meaningful if and only if it relies on prior knowledge about underlying mechanism and specific prerequisites that are relevant for a particular phenomenon/effect. Especially when investigating multifunctional proteins like ALDH1A3, which is involved in various cellular processes (metabolism, differentiation, DNA repair, proliferation) and exerts a myriad of effects that can vary widely depending on the type of cells and molecular context. Our study provides first evidence that a widespread view that ALDH1A3 promotes self-renewal of undifferentiated GSCs is not a universal rule. Our principal finding is that this is not always the case and that ALDH1A3 can influence proliferation at the onset of differentiation in differentiation-capable GSCs and inhibit self-renewal in the context of radiation response. Our findings provide new clues and contribute to the conceptual continuity in further elucidation of ALDH1A3 actions in glioblastoma promotion and radioresistance. While completely agreeing with the reviewer that novel aspects of ALDH1A3 actions uncovered in our study need to be validated in other experimental models we would like to point out that such validation cannot be done in just any other cell line but would require GSC lines that fulfil specific requirements. Of note, many glioblastoma cell lines widely used in previous research on ALDH1A3 do not fulfil important criteria for GSCs such as ability to transit between different cell states. For that very reason, the relationship between ALDH1A3 activities and cell state transitions uncovered through our investigations has been overlooked in previous studies. In our study, the association between ALDH1A3 activities and cell state were uncovered in p53-deficient GSCs that belong to the proneural subtype, which is consistent with the hypothesis that ALDH1A3 promotes glioblastoma by acting primarily in proneural GSCs. Given the molecular and functional diversity of GSCs constituted by distinct subtypes a further validation of the effects revealed in our study with proneural GSCs would require prior characterization of different GSC lines with respect to the molecular subtype, degree of inherent plasticity, tumour-initiating capacity and radioresistance. Furthermore, the association between ALDH1A3 effects and radiation necessitates prior knowledge about functional status of p53, which is a universal master regulator of radiation response. Establishing all these characteristics is on its own an independent investigation that would be required for effective validation of the hypothesis proposed in our study.”

Reviewer: Fig.2 shows that GSC 1095 has intermedium level of ALH1A3 expression. Therefore, this cell line can be used for overexpression and knockdown or knockout. Additional cell lines are also available to use in overexpression/KD/KO studies.

Response: As explained above and in our previous response, the decisive factor is not the mere availability of additional cell lines but suitability requirements including functional status of p53, self-renewal/differentiation potential and molecular subtype.

Reviewer: Analysis of p53 status should not be difficult to do.

Response: Analysis of the functional p53 status in different cell lines is not only a time consuming task that is impossible to complete within the timeframe of manuscript revision, it was also beyond the scope of this study. 

Reviewer: Inability to generate cells with sufficient target KD is not an excuse.

Response: In our previous response, we have provided a detailed explanation for why we decided not to pursue ALDH1A3 silencing that proved unsuccessful in our hands but focus on ALDH1A3 overexpression as an approach to model clinically and biologically relevant features of GSCs. As our explanations have been uncommented by the reviewer, we provide them here again: “Tumour-promoting effects of ALDH1A3 are associated with its overexpression not downregulation or loss. Therefore, ectopic overexpression in GSCs with low levels of endogenous ALDH1A3 is a straightforward approach to model the real-life situation. For that reason, the p53-null line GSC-1095, which has very low levels of endogenous ALDH1A3 was chosen for overexpression experiments. Parallel knockdown experiments with GSC-1095 line were also performed but they proved to be sufficiently informative because of marginal difference in the levels of ALDH1A3 in cells transduced with ALDH1A3-shRNAs or scrambled-shRNA. We therefore focused on ALDH1A3-overexpressing models to reproduce the phenomenon of ALDH1A3 overexpression in glioblastoma”.

Reviewer: Authors did not show if target expression levels correlate with tumor grade or subtype or survival. Authors found excuse not to perform in vivo experiment.

Response: In our previous response, we have provided a detailed explanation for why performing survival analyses is impossible within the timeframe given for the manuscript revision, namely: “Performing survival analyses within the timeframe for manuscript revision is an unrealistic goal because the average survival time for GSC-1095 xenografts is around 300 days (Müller et al 2023; https://www.mdpi.com/2073-4409/12/9/1290). To compensate for this limitation, we made use of surrogate tests for tumorigenicity, namely the colony formation and growth rate assays. The results shown in a new Figure 10 corroborate our hypothesis that ALDH1A3 boosts proliferation and tumorigenic potential in both naiv and radiation-treated differentiated progenies.” Unfortunately, our explanation has not been commented. Neither does the reviewer provide any comments on the results of new experiments, which have been conducted to model in vivo outcomes. The notion that our explanations serve the purpose of finding some kind of “excuse” might have been more helpful if accompanied by concrete comments/critiques/suggestions.

Reviewer: Currently characterized and fast growing GSC lines available from different laboratories.

Response: As explained above, the decisive factor is not the mere availability of additional cell lines but suitability requirements including functional status of p53, degree of stemness and molecular subtype. We would like to point out that “fast growing GSC lines” mentioned by the reviewer are likely to be differentiated progenitors rather than undifferentiated GSCs, which are slow-cycling cells that divide rarely.

Reviewer: Authors corrected only “Other comments” section of the review.

Response: That is because “Other comments” were the only concrete comments related to the manuscript content.

Reviewer 2 Report (Previous Reviewer 4)

Comments and Suggestions for Authors

Authors have improved the paper.

Author Response

Reviewer: Authors have improved the paper.

Our response: Thank you very much for your time and efforts to help improving our manuscript! 

Reviewer 3 Report (Previous Reviewer 2)

Comments and Suggestions for Authors

Müller et al. used GSC cell cultures to investigate the role of the ALDH1A3 gene in stemness and radioresistance in glioblastoma. 

1. What are the statistical methods used in graphs in Fig. 1?

2. Include background information on GSC-1095 in Fig. 2 corresponding paragraph.

3. Fig. 4A, please use empty/solid arrowheads to indicate different types of cells and include quantification for these images. 

4. Fig. 4B, what's the marker here?

5. Fig. 5A, pool #3 bands too faint

6. Fig. 9A&B, include markers of fluorescent here

7. Fig. 9C, what statistical method is used?

Comments on the Quality of English Language

N/A

Author Response

1. What are the statistical methods used in graphs in Fig. 1?

Response: relative proportions of proteins (pixel intensity) were determined by using the Excel “CountIf” function.

2. Include background information on GSC-1095 in Fig. 2 corresponding paragraph.

Response: background information on GSC-1095 and reference to previous study describing clinically relevant characteristics of this line has now been provided in Legend to Fig. 2 (lanes 203-205).

3. Fig. 4A, please use empty/solid arrowheads to indicate different types of cells and include quantification for these images.

Response: representative undifferentiated (GFAP-) and differentiated (GFAP+) cells are indicated by solid or empty arrowheads, respectively. Numbers of GFAP-/DAPI+ and GFAP+/DAPI+ cells are indicated in the table below the image. Explanation is provided in the legend to Fig. 4A (lines 240-242).

4. Fig. 4B, what's the marker here?

Response: Fig. 4B shows immunohistochemical staining for a putative neural stem cells marker nestin using antibody specific for human nestin. This information is provided in the legend to Fig 4B. 

5. Fig. 5A, pool #3 bands too faint

Response: Fig. 5A shows the results of testing of three different pools of clones generated by GSCs transduction with ALDH1A3-coding lentivirus. Such testing is important to ensure the efficacy of transgenes expression. From the three pools tested, lentivirus-mediated overexpression of ALDH1A3 proved to be efficient in pools #1 and 2 but not in pool#3. That is why bands corresponding to ALDH1A3-specific transcripts expressed in pool #3 are faint. 

6. Fig. 9A&B, include markers of fluorescent here

Response: fluorescence markers are now indicated in the legend to Fig. 9A,B.

7. Fig. 9C, what statistical method is used?

Response: relative proportions of GFAP-/HSP3S10+ and GFAP+/H3S10+ cells were determined by using the Excel “CountIf” function. Statistical data analysis was performed by using the Student’s t-test and presented as mean ± SD. A P value <0.05 was considered statistically significant. Information concerning statistical methods is provided in Materials and Methods.

Round 2

Reviewer 3 Report (Previous Reviewer 2)

Comments and Suggestions for Authors

Most of the concerns I mentioned in the last revision have been addressed.

Please put all the immunostaining/immunofluorescence markers together in the images, not only in the figure legends, that would be more reader-friendly.

Also the arrow/arrowheads are missing which I've mentioned in the last revision and you responded.

Comments on the Quality of English Language

N/A

Author Response

Please put all the immunostaining/immunofluorescence markers together in the images, not only in the figure legends, that would be more reader-friendly. >>>Immunostaining/immunofluorescence markers have now been indicated in the corresponding images. 

Also the arrow/arrowheads are missing which I've mentioned in the last revision and you responded.>>> We sincerely apologize for overseeing that Fig. 4 in the previously submitted revised version has accidently been “replaced” with the same old version. In the current revised version, Fig. 4 modified in accordance with the reviewer’s suggestion is now shown.

This manuscript is a resubmission of an earlier submission. The following is a list of the peer review reports and author responses from that submission.

Round 1

Reviewer 1 Report

Comments and Suggestions for Authors

Muller et al.

ALDH1A3 contributes to radiation-induced inhibition of self-renewal and promotes proliferative activity of p53 deficient glioblastoma stem cells at the onset of differentiation

This study investigates the role ofALDH1A3 on GSC proliferation, differentiation and stemness. This study is limited to single GSC line which is exogenously overexpress ALDH1A3. Experiments with overexpression and knockdown are recommended. Without additional cell lines conclusions made by this study are rather speculative since reproducibility of observed effects are not demonstrated. Inclusion of animals (XRT vs untreated) and subsequent analysis of survival and stemness marker expression is recommended to demonstrate clinical relevance.

Other comments: 

Lines: 130-131: one of vector names is missing insert ALDH1A3.

Lines 145-149 a variety of antibodies are listed, but no result are shown in subsequent result section using these antibodies. 

 Fig.5 shows 2 clones of cells after viral transduction. Seems that advantage of lentiviral transduction and utilization of pool of clones from heterogenous GSC line is not used. However, in subsequent figures no individual clones are mentioned. 

Reviewer 2 Report

Comments and Suggestions for Authors

Muller et al. used glioblastoma stem cells to identify the role of ALDH1A3 in regulating stemness and radioresistance. They found that this impact is cell state-dependent, which is more significant within differentiated GSC progenies. 

1. Scale bars are missing in Fig.4 and 9. 

2. Fig. 1, do pixel intensity measurements for quantification (relative to control load), categorize them by cell types, would be easier and for readers to see the heterologous expression of different proteins (similar to the graph in Fig.2). 

3. Fig. 4A, you need higher magnification for cell images, in order to see the overlapping of GFAP and nuclei staining in individual cells level. 

4. Fig. 4B, consider including 1. surgical site dissection image 2. xenograft BF image under dissection microscope 3. Nuclei staining is too faint, would suggest redoing it 4. what's the correlation between the bottom two images? they look morphologically different and with different magnifications. 

5. Fig. 9A, you need 3 channels merged image here (DAPI+H3S10+GFAP), and I would suggest finding a more representative image that contains more triple-labeled cells. 

6. similar to comment 5, you need 3 channels of merged image. Also I think you need quantification of the percentage of single/double/triple-labeled cells for what you try to show in Fig. 9. 

Comments on the Quality of English Language

Overall writing is good, but some long sentences are hard to read, especially with the various cell lines you have incorporated into the research. Maybe working with a professional academic writer to improve the precision and flow. 

Reviewer 3 Report

Comments and Suggestions for Authors

In this study, the authors report that ALDH1A3 isoform may contribute to glioma proliferation under radiation by restraining self-renewal. The overall study solely relies on one ALDH1A3 overexpression isogenic clone, which may miss the heterogeneity associated with tumorigenesis, and may generate overexpression associated artifacts. Thus, at least loss of function studies are necessary to confirm observed phenotypes. In addition, the rationale to further increase the ALDH1A3 expression in an ALDH1A3-high cell line remains unclear. How ALDH1A3 may mediate observed phenotype is not examined. This study is currently at its preliminary stage and more in-depth investigations would be needed prior to its consideration for publication.

The current format of the manuscript is hard to follow. More detailed descriptions for the experiments would be needed- eg Fig. 1 are these patient samples or cell lines (human or mouse)?

Fig. 1: How about other ALDH isoforms? The rationale to only examine ALDH1A3 is weak.

Fig. 5: The cellular localization of GFP-ALDH1A3 would need to be examined.

Comments on the Quality of English Language

Would need to be improved.

Reviewer 4 Report

Comments and Suggestions for Authors

In my opinion, the article entitled ALDH1A3 contributes to radiation-induced inhibition of self-2 renewal and promotes proliferative activity of p53 deficient glioblastoma stem cells at the onset of differentiation is a good work and it is suitable for publication in Cells with minor/major revision. The results are very well showed, documented and the text is properly written.

As I have said, it is a good article, and authors seem to have worked hardly. They have developed many different techniques to try to elucidate the role of ALDH1A3 on irradiated cell proliferation status under pro and no differentiation conditions. However, I have some suggestions:

Minor points:

Material and Methods

1.- In western blot paragraph authors have described many antibodies that you have not used, or at least, you have not showed in the paper results. When have you used p53, AKT, LC3B, etc….

2.- You should explain what ELDA exactly is.

Results

1.- CD133, PDGFRA and PDGFRB antibodies are not included in wester blot section.

2.- ndGB and recGB acronyms should be previously reported.

3.- Figure 2: Why haven´t you made the correlation ALDH1A3/PDGFRB levels? It is obvious that they are also inversely correlated.

4.- I do not understand how ELDA works. Authors must explain how it works in order to better understand results.

5.- Why haven´t you showed ELDA results for U87 cells? It would help to understand ELDA data if they do not self-renew but differentiate.

6.- What are you exactly staining with GFAP?

7.- Line 209: one and lefts over

8.- Line 275-276: both cannot be 1095ALDHA1A3+

In my opinion, if you think that irradiated ALDHA1A3+cells are boosted to proliferate, instead of differentiating, you should perform clonogenic assays in order to be sure that differentiated progenies have an active role in glioblastoma propagation.